# A cross-cultural examination of temporal orientation through everyday language on social media

Xin Daphne Hou[1]*, Sharath Chandra Guntuku[2]*, Young-Min Cho[2], Garrick Sherman[2], Tingdan Zhang[3], Mingyang Li[2], Lyle Ungar[2], Louis Tay[1]

**1** Department of Psychological Sciences, Purdue University, West Lafayette, IN, United States of America, **2** Department of Computer and Information Science, University of Pennsylvania, Philadelphia, PA, United States of America, **3** Collaborative Innovation Center of Assessment Toward Basic Education Quality, Beijing Normal University, Beijing, China

* hou12@purdue.edu (XDH); sharathg@seas.upenn.edu (SCG)

## Abstract

Past research has shown that culture can form and shape our temporal orientation–the relative emphasis on the past, present, or future. However, there are mixed findings on how temporal orientations vary between North American and East Asian cultures due to the limitations of survey methodology and sampling. In this study, we applied an inductive approach and leveraged big data and natural language processing between two popular social media platforms–Twitter and Weibo–to assess the similarities and differences in temporal orientation in the United States of America and China, respectively. We first established predictive models from annotation data and used them to classify a larger set of English Twitter sentences ($N_{TW}$ = 1,549,136) and a larger set of Chinese Weibo sentences ($N_{WB}$ = 95,181) into four temporal catetories–past, future, atemporal present, and temporal present. Results show that there is no significant difference between Twitter and Weibo on past or future orientations; the large temporal orientation difference between North Americans and Chinese derives from their different prevailing focus on atemporal (e.g., facts, ideas) present (Twitter) or temporal present (e.g., the "here" and "now") (Weibo). Our findings contribute to the debate on cultural differences in temporal orientations with new perspectives following a new methodological approach. The study's implications call for a reevaluation of how temporal orientation is measured in cross-cultural studies, emphasizing the use of large-scale language data and acknowledging the atemporal present category. Understanding temporal orientations can guide effective cross-cultural communication strategies to tailor approaches for different audience based on temporal orientations, enhancing intercultural understanding and engagement.

## Introduction

"We are not makers of history. We are made by history."–*Martin Luther King, Jr.*

**Data Availability Statement:** Data used for this study is available at Github repository [https://github.com/JeffreyCh0/tw_wb_temporal/tree/main]. The annotation processes are fully captured

in two written annotation guidelines in English and Chinese, published on Open Science Framework [https://osf.io/5qy4e/?view_only=5d23e55e25a24421a6fb93f7d88ba0b8].

**Funding:** The author(s) received no specific funding for this work.

**Competing interests:** The authors have declared that no competing interests exist.

"Those who do not plan for the future will find trouble at their doorstep."–*Confucius*

Time profoundly shapes human societies, from everyday activities to philosophical pursuits of meanings of life. Meanwhile, time perspectives—how we view time—also pervade our lives and can influence how we process circumstances, make decisions, and give different meanings to our lives [1–4]. Even beyond individual outcomes, it is believed that fundamental differences in time perspectives can shape interhuman and intersocietal interactions and conflicts. Within psychology, one specific type of time perspective is temporal orientation, which describes how much people direct their attention to events in different time points (i.e., past, present, or future) [1–4]. Current work has mainly focused on three dimensions of temporal orientations—past, present, and future—and found that individual's attention allocated to different timepoints can have impactful effects, such as life satisfaction [5], mental health [6], and social responsibilities [7].

For example, trauma survivors with past temporal orientation experience elevated levels of distress even when controlling for the degree of ruminations; organizational studies have also shown that having a past temporal orientation is related to turnover decisions after a pay cut [1]. By contrast, future-oriented individuals are more likely to set goals [8], form healthy eating habits [9], make responsible financial decisions [4, 10], care about social and environmental issues [7], and support future-oriented policies [11].

## Temporal orientation and culture

Temporal orientation is shaped by the culture in various ways, including the length of history, the prevalent value system, and the dominant language of the culture [1, 11–14]. Collectively considering these cultural factors, a commonly accepted view is that East Asian cultures have a greater past orientation due to longer histories and greater values in traditions, whereas North American culture is considered more future-oriented [14–18]. Moreover, North Americans are considered to be more present-oriented than Chinese, such that they live more in the here and now than the past or future [16, 19]. Nevertheless, the evidence so far has been mixed. Some scholars have debated with evidence that Chinese are more future-oriented than North Americans [20, 21] due to the long-term orientation in East Asian countries [16, 22] and their predominant futureless language–Mandarin Chinese [12, 13]. Indeed, 71.43% of Chinese participants reported the highest tendency to think most about the future, whereas only 43.33% of Americans reported the same tendency [15].

From a linguistic perspective, the dominant language spoken in the country can predispose its speakers' temporal orientation based on the use of time tenses [12, 13, 23]. A futured language like English adopts time-tense and auxiliary verbs (e.g., "will"), which guide speakers to think of the future as a separate event from the present and past. In contrast, a futureless language (i.e., Chinese) does not require a time tense when discussing past, present, or future. Therefore, Chinese speakers might be expected to view the future as more integrated with present and past and are believed to engage in more future-oriented thoughts and behaviors [20, 21, 24]. These mixed findings are further complicated because studies have found that people are often not thinking about past, future, or the present in both the American (11–22%) and Chinese contexts (22.02%) [25, 26]. Rather, there may be a fourth category of atemporal present (e.g., facts, ideas).

Some of the mixed findings may be due to the sampling and methodological challenges. Sampling-wise, most past cross-cultural comparisons between East Asians and North Americans have relied on undergraduate participants and Asian Americans representing East Asian cultures with limited sample sizes (average of 120 North American participants and 179 East

Asians) (S1 Table), making it difficult to generalize. Methodologically, cross-cultural comparisons of temporal orientation have relied primarily on self-report scales on which there are concerns of comparability when translating English scales to other language contexts [27, 28]. The translated version of the temporal orientation scales has reported modest levels of internal consistency reliability ($< 0.5$–0.77) [27, 29, 30]. Relatedly, existing cross-cultural studies have rarely conducted equivalence analyses when comparing temporal orientations between cultures [28]; some studies including a meta-analysis have reported poor face validity, reliability, structural validity and equivalence of existing time perspective models and measurements [28, 29, 31, 32]. Part of these issues reside in the lack of underlying general theoretical basis and construct clarity [28] that leads to measuring a mix of cognitive, affective, and behavioral-based temporal focus [2]. Critically, the use of self-report scales may limit the possible categories of time orientation to only past, present, and future, when researchers have found a possible atemporal present category as well [26].

## Purpose of the present study

To address the mix findings and methodological challenges, in this study, we aim to examine the temporal differences between North American and Chinese cultures using naturally occurring social media language, as a behavior-based measure of temporal orientation and seek to discover new knowledge through big data and natural language processing. Language use on social media provides a rich source of behavioral data in original language contexts, from which psychological characteristics can be predicted through machine learning models while ensuring high ecological validity [33–35]. Our study focuses on comparing two popular social media platforms–Twitter (U.S.) and Sina Weibo (China)–representing North American and East Asian cultures, respectively. In addition, instead of the traditional three categories of temporal orientations–past, present, and future, we further explored the subcategory of "atemporal present" in line with previous studies [25, 26]. We define the atemporal present category as one that does not have a specific time of reference that a person directs their attention to; for example, it can be a generic description of an object (e.g., "apples are red"), a habit (e.g., "I exercise in the morning"), or a logic/judgement (e.g., "my family makes me happy" or "football is fun"). On the other hand, the traditional temporal present orientation describes a clear and strong focus on the here and now (e.g., "I'm having a good time at the party").

## Materials and method

This study was reviewed and approved by the Institutional Review Board of University of Pennsylvania (#816091). Following a mixed method, we developed two predictive models based on two sets of annotated posts on Twitter and Weibo and applied the models to two larger sets of social media text data to automatically identify the temporal orientations of the posts. The posts data were analyzed at the sentence level meaning that each post was broken down into sentences and each sentence was analyzed individually. We chose to focus on the sentence level to reduce ambiguity as one post can include a mix of sentences describing different time points. The following steps were followed and described in detail below: a) obtaining a set of sample posts from both platforms; b) annotating the temporal orientations of the posts at the sentence level by native speakers; c) establishing and applying the temporal predictive models.

### Sampling social media data

**Data collection.** We use public written messages posted on Weibo and Twitter as our dataset. For Twitter, we use Qualtrics to compensate participants to share their demographic information and Twitter handles in the survey. Then we use informed consent to collect their

Twitter posts. In our dataset, there are 3,133 users with around 3.6 million posts originally posted from 2007 to 2016. Since Weibo does not provide an API tool for post collection in a time window, we used breadth-first search on a random set of users from a public dataset to gather Weibo posts. Our data contains 29 million posts from 859,054 users from 2014, and their demographic information is based on self-reported profiles. We also dropped users with less than 500 words across their posts, unusual age ($>/= 100$ years) and other genders except for male and female, for obtaining reliable psychological estimates [43, 44]. Finally, our dataset has 668,257 Weibo posts from 8,731 users. Data used for this study is available at Github repository (https://github.com/JeffreyCh0/tw_wb_temporal/tree/main) with all the features needed to replicate this study.

**Language preprocessing.**  Re-tweets and Shares are removed from both datasets ('RT @USERNAME:' on Twitter and '@USERNAME//' on Weibo). For tokenizing, we used THU-LAC [45] for Weibo posts and happierfuntokenizing [46] for Twitter. We also dropped uncommon words and phrases by removing words used by fewer than 1% of users, which is helpful for our model to be generalized to edge cases.

**Annotation data selection.**  Twitter posts were randomly sampled for annotation from the 1% sampled stream of tweets. This random stream of public tweets comprises approximately 1% of the total Twitter posting volume. 3,000 tweets were sampled from the 1% stream for annotation. Weibo data was also randomly sampled from our full corpus of Weibo posts. We sampled 3,000 Weibo posts to keep the size of the data sets consistent.

## Annotating temporal orientations

We followed the steps depicted in Fig 1 to annotate the sampled data in preparation for building the predictive models. The annotation processes are fully captured in two written annotation guidelines in English and Chinese, published on Open Science Framework (https://osf.io/5qy4e/?view_only = 5d23e55e25a24421a6fb93f7d88ba0b8) for future research replications. Some of the key rules include: a message referring to the immediate present within an hour would be coded as 0 and non-interpretable messages are coded as "NA". Messages that were coded as "NA" by all three raters were excluded.

Three bilingual coders annotated one training set of posts at sentence level from social media posts collected on each platform ($N_{TW} = 2,135$, $N_{WB} = 3,593$) from demographically similar users (67.32% female on Twitter and 69.04% female on Weibo; $Age_{TW} = 27.6$, $Age_{WB} = 25.76$). For Twitter, we continued the annotation created by Park et al. [33], who initially coded for only three dimensions–past, present, and future. Based on the sentences labeled as "present" from their work, three annotators re-coded the "present" sentences between "temporal present" and "atemporal present". One of the key distinguishers between "temporal present" and "atemporal present" is whether the sentence describes or states something that is factual or perpetually true. The intraclass correlation coefficient (ICC) was used to evaluate the accuracy of the codings. ICCs for English Twitter annotations were 0.78 (past), 0.87 (future), 0.83 (atemporal present), and 0.83 (temporal present).

For Weibo, we asked three native speaker annotators to label sentences between "past", "temporal present", "atemporal present", and "future". Due to the language grammatical ambiguity of temporal orientation, we allowed multilabel annotations in Weibo posts. We first classified whether the Weibo post was a personal post or a non-personal post; non-personal posts including advertisements, news, quotes, and the like were excluded. We then coded the personal posts in terms of temporal orientation at the sentence level. A frame of reference training was provided prior to the annotation work, in which all annotators became familiar with the annotation guideline and practiced 50 Weibo posts. Each annotator then annotated the same

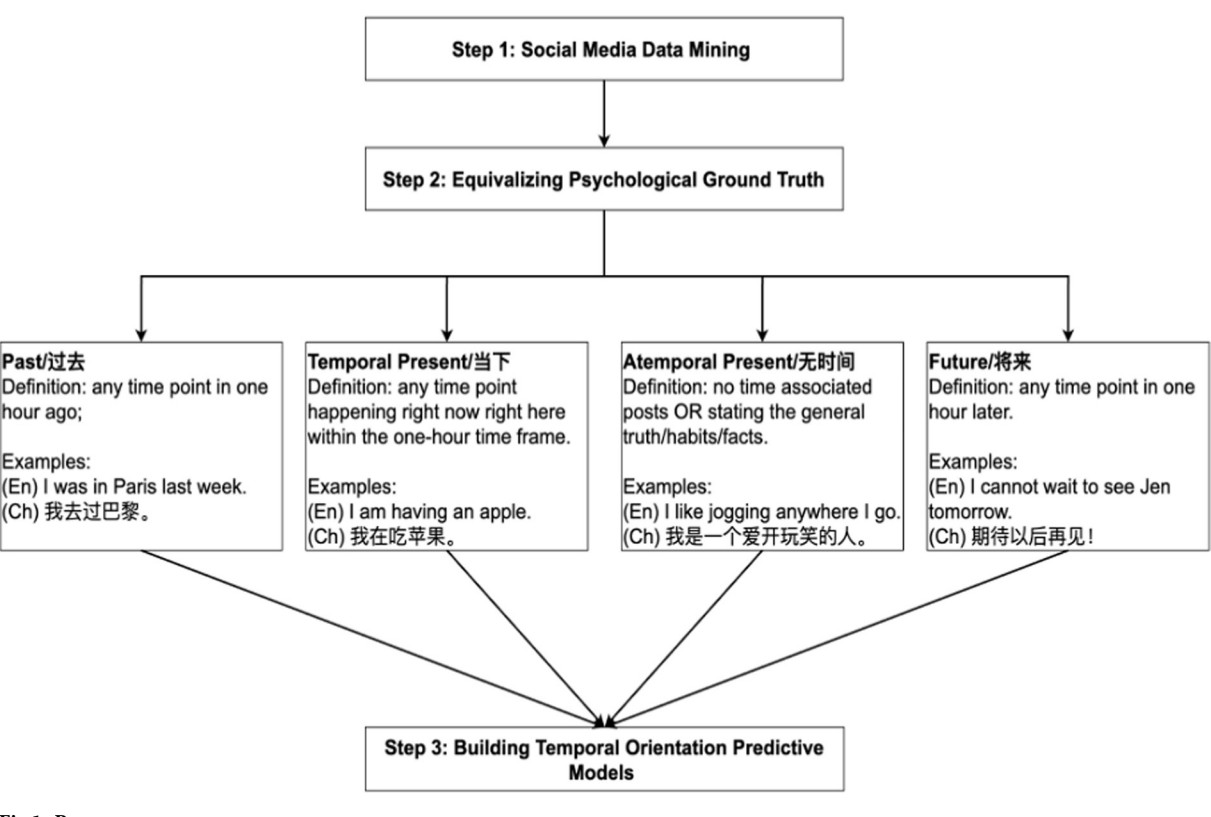

**Fig 1. Process map.**

set of 6891 sentences from 2973 Weibo posts. ICCs for Chinese Weibo annotations were 0.79 (past), 0.86 (future), 0.72 (atemporal present), and 0.74 (temporal present).

## Building predictive models

We extracted normalized frequencies of words and phrases (1-3grams) and Linguistic Inquiry Word Count (LIWC-2015) for each message. We used these as input features in a logistic regression classification model: TemporialOrientation = $B_0$ + Sum($B_i$*LangFeat$_i$), where Lang-Feat$_i$ is each of the language features along with a feature selection pipeline that uses a univariate feature selection (with a family wise error rate of 60) and Randomized Principle Component Aanalysis [36] to reduce the dimensions from the scikit-learn library and predict dummy variables representing whether each message was annotated as past, present, and future. Lastly, the predictive models classified a larger set of English Twitter sentences ($N_{TW}$ = 1,549,136) and a larger set of Chinese Weibo sentences ($N_{WB}$ = 95,181) into the four categories–past, future, atemporal present, and temporal present, after eliminating non-personal posts. Chi-square analyses were applied to compare the differences between Twitter and Weibo in each of the four categories. Similar methods have been used in prior works to predict psychological constructs such as personality [37], stress [38], politeness [39], as well as prior work on predicting temporal orientation within the U.S..

## Results

The annotations resulting in satisfactory classification accuracy (mean $ICC_{TW}$ = 0.83; mean $ICC_{TW}$ = 0.78) served as the psychological ground truth for building the predictive models.

Then we applied the predictive models to a larger sample of social media sentences written in the original languages using user-matched datasets ($N_{user}$ = 2,191 for both Twitter and Weibo) and predicted the temporal orientations of each sentence. Measuring the frequency of time-points minimizes the reliance on self-reports and allows us to parcel out temporal orientations–how often people think about an event in the past, present, and future from temporal values–how much value people attach to an event in the past, present, and future [18]. In addition, we visualized the predicted results using word clouds so that both expert and lay audiences can view a summary of the results [40] (Fig 2). The highly correlated words to each temporal orientation (i.e., "was"/ "" for past, "tomorrow"/ "" for future, "now" for "temporal present") add to the classification accuracy evidence beyond the satisfactory ICC levels.

Results from the larger testing set of social media sentences agreed with results from the initial training sets, which were used to build the predictive models. Overall, there is a significant temporal orientation difference between English Twitter and Chinese Weibo ($\chi^2$ = 398,998, $p$ < .001, $V$ = 0.49) (Tables 1 and 2). When we exclude "present" (i.e., temporal present and atemporal present) to only compare past and future (Table 3), the results showed only a small difference in past and future between English Twitter and Chinese Weibo ($\chi^2$ = 56.190, $p$ < .001, $V$ = .01) (Table 4). Although the differences between past and future are significant, Cramer's V indicates a rather weak association between these two platforms and temporal orientations towards past or future.

Shifting the focus to "present" only (Table 5), Twitter and Weibo showed significant differences ($\chi^2$ = 196,549.15, $p$ < 0.001, $V$ = .46) (Table 6). What drives this significant difference is the large number of "temporal present" posts in Chinese Weibo (89.6%) and a large number of "atemporal" posts in English Tweets (41.5%). Both the training and testing sets confirmed a substantial atemporal present posts in addition to past, present, and future on both platforms. When considering "temporal present" alone, Chinese Weibo was predominantly "temporal present" (91.9% vs. 22.1% on Twitter), while English Twitter was predominantly "atemporal present" (77.9% vs. 8.1% on Weibo). In summary, the significant differences in "temporal present" and "atemporal present" primarily drive the overall temporal differences between English Twitter and Chinese Weibo, rather than differences in past and future orientations.

## Discussion

Previous studies have generally posited temporal differences in past and future between Eastern and Western cultures [14–17, 19–21]. Nevertheless, although our study showed a significant temporal difference in past and future between U.S. and China, the effect size was negligible. Instead of a significant difference between past and future, atemporal versus temporal presents have revealed the biggest difference between U.S. and China, such that Chinese Weibo posts focus more on the "here" and "now" (temporal present) while English Tweets focus more on factual information (atemporal present).

One plausible explanation for the small effect size between past and future orientations is that previous studies that have adopted self-report scales have measured temporal orientation as a mix of cognitions, affects, and behaviors [2]. Such operationalizations measure different temporal constructs on one scale, including, for example, temporal values and beliefs (i.e., "it's more important for me to enjoy what I am doing" and "I believe it is important to save for a rainy day" [6]. By contrast, we believe that our approach that draws on large-scale language to estimate temporal orientation is more accurate in determining what people direct their attention to behaviorally. Although the mainstream notion that North Americans *value* the future more than the past while Chinese value the past more than the future may be true [17], it appears that the frequency of thinking about them is a separate story. This finding also suggests

|  | Weibo | Twitter |
|---|---|---|
| **Past** | | |
| **Future** | | |
| **Temporal Present** | | |
| **Atemporal Present** | | |

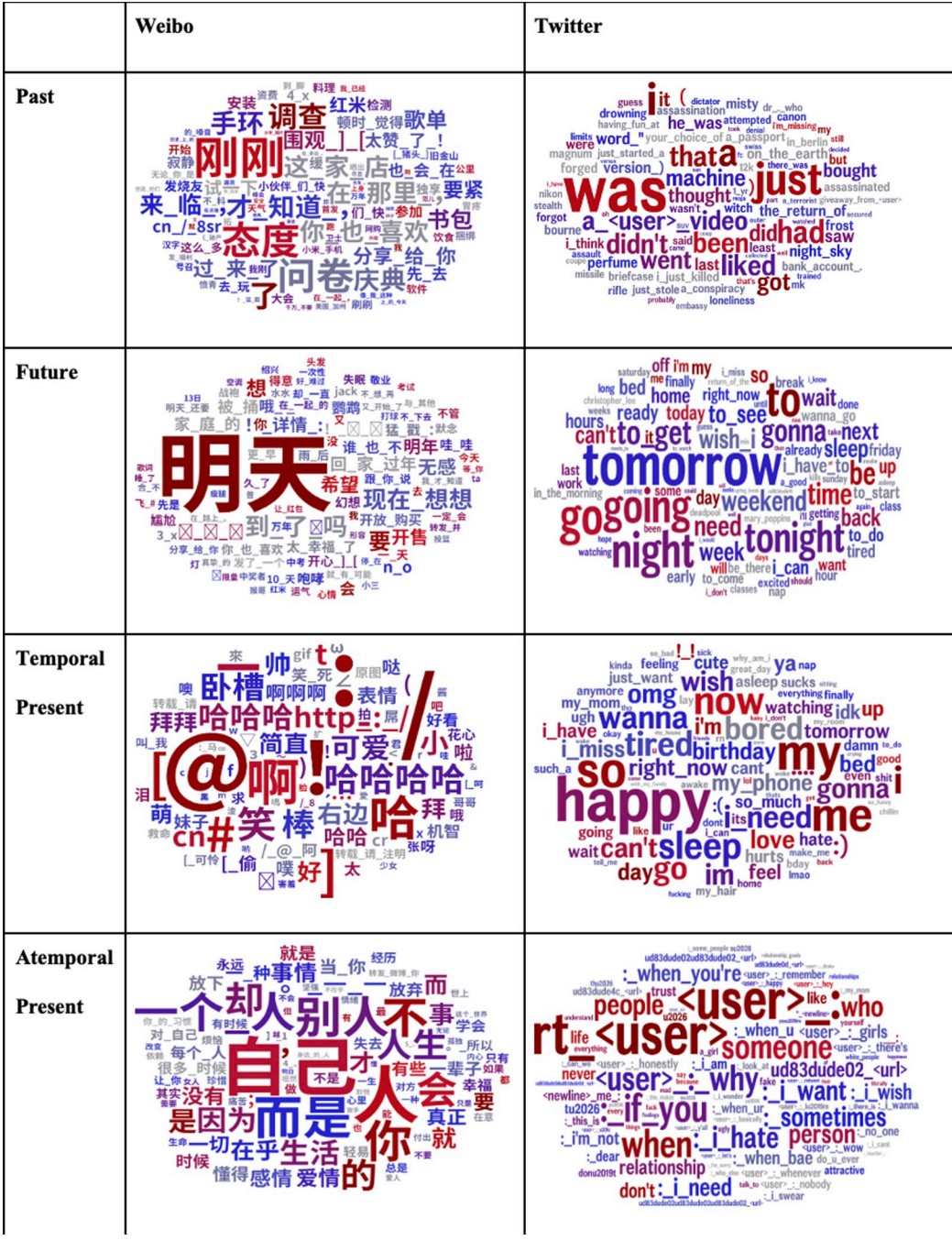

**Fig 2. Wordclouds results.**

that language use on social media is largely present focused on both temporal present and atemporal present, converging with the 50% and 61% present focus (in English context) found by two other studies that adopted similar approaches using social media and machine learning models [33, 41]. To our knowledge, there has been no previous temporal orientation study done using Chinese social media data.

**Table 1. Temporal orientation crosstabulation.**

| | | | Atemporal Present | Future | Past | Temporal Present | Total |
|---|---|---|---|---|---|---|---|
| Platform | Twitter | Count | 642406 | 305597 | 418833 | 182300 | 1549136 |
| | | Expected Count | 612331.5 | 289017.4 | 395707.2 | 252079.9 | 1549136.0 |
| | | % within Platform | 41.5% | 19.7% | 27.0% | 11.8% | 100.0% |
| | | Adjusted Residual | 205.4 | 142.1 | 177.1 | -631.3 | |
| | Weibo | Count | 7548 | 1178 | 1187 | 85268 | 95181 |
| | | Expected Count | 37622.5 | 17757.6 | 24312.8 | 15488.1 | 95181.0 |
| | | % within Platform | 7.9% | 1.2% | 1.2% | 89.6% | 100.0% |
| | | Adjusted Residual | -205.4 | -142.1 | -177.1 | 631.3 | |
| Total | | Count | 649954 | 306775 | 420020 | 267568 | 1644317 |
| | | Expected Count | 649954.0 | 306775.0 | 420020.0 | 267568.0 | 1644317.0 |
| | | % within Platform | 39.5% | 18.7% | 25.5% | 16.3% | 100.0% |

Additionally, the atemporal present category has emerged to be salient on both English Twitter and Chinese Weibo. The atemporal present category has been often overlooked and understudied yet consistently identified in previous studies (e.g., cluster analysis) [2, 4]. For example, across mind-wandering studies, the proportion of atemporal present focus varies between 11–22% of reported temporal focus [25, 26, 42–44], yet this option is often not present on the existing temporal measurement instruments [2]. Our study provides strong evidence to support the existence of the atemporal category. Overlooking this category of temporal orientation would lead to a biased understanding of people's temporal orientations. Indeed, in mind-wandering research, Jackson et al. found that atemporal present response option was used at least as frequently as past or future; however, when atemporal response was not presented as an option, there was a significant amount of future thinking indicated by participants [26]. In other words, when no atemporal present option was included in a survey measure, participants' prospective bias may have been induced such that they tend to classify atemporal present thoughts as future or past.

Last, temporal present has a dominating presence on Chinese Weibo, whereas atemporal present had the highest presence on English Twitter. According to the construal level theory, East Asians have a more concrete and contextual mental representation of the future compared with North Americans [19, 45], leading Chinese to pay attention to more contextual information (i.e., past and future) as backgrounds for the temporal present [19]. Similarly, East Asians tend to take a more holistic approach by attending to the entire field in reasoning, whereas Westerners are more analytic, drawing upon facts, objects, and formal logic to understand the world [46]. These unique cultural factors provide plausible explanations for this temporal difference in temporal present and atemporal present. A holistic thinking or more concrete mental

**Table 2. Chi-square test results for all temporal orientation dimensions.**

| | Value | df | Asymp Significance (2 sided) |
|---|---|---|---|
| Pearson Chi-Square | 398997.90[a] | 3 | .000 |
| Likelihood Ratio | 278186.86 | 3 | .000 |
| Phi | .49 | | .000 |
| Cramer's V | .49 | | .000 |
| N of Valid Cases | 1644317 | | |

a. 0 cells (0.0%) have expected count less than 5. The minimum expected count is 15488.13.

**Table 3. Past and future orientation crosstabulation.**

| | | | Future | Past | Total |
|---|---|---|---|---|---|
| Platform | Twitter | Count | 305597 | 418833 | 724430 |
| | | Expected Count | 305776.8 | 418653.2 | 724430.0 |
| | | % within Platform Adjusted | 42.2% | 57.8% | 100.0% |
| | | Residual | -7.5 | 7.5 | |
| | Weibo | Count | 1178 | 1187 | 2365 |
| | | Expected Count | 998.2 | 1366.8 | 2365.0 |
| | | % within Platform Adjusted | 49.8% | 50.2% | 100.0% |
| | | Residual | 7.5 | -7.5 | |
| Total | | Count | 306775 | 420020 | 726795 |
| | | Expected Count | 306775.0 | 420020.0 | 726795.0 |
| | | % within Platform | 42.2% | 57.8% | 100.0% |

representation of the future/past can influence Chinese speakers' tendency to use present language in describing a future or past event. On the contrary, Westerners post more factual statements on Twitter, reflecting on their preferences for objects, facts, and logic.

## Implications

Implications of the study come in several folds. First, and formost, the study highlights the need to reconsider the measurement of temporal orientation in cross-cultural studies. Previous studies have often relied on self-report scales that capture a mix of cognitions, affects, and behaviors related to past, present and future orientations. However, the study suggests that using large-scale language data can provide a more accurate understanding of what individuals actually direct their attention to behaviorally. In addition, the study emphasizes the salience of the atemporal present category, which has been consistently identified but often overlooked in previous research. Researchers should acknowledge and include the atemporal present category in temporal measurement instruments to ensure a comprehensive assessment of temporal orientations.

Furthermore, the insights gained from this study can have practical implications for cross-cultural communication. Understanding the temporal orientations of different cultures can aid in developing more effective communication strategies. For instance, if Chinese speakers have a dominant presence of temporal present on Weibo, it might be valuable to consider contextual information, such as past and future, when communicating with this audience.

**Table 4. Chi-square test results for past and future orientations.**

| | Value | df | Asymptotic Significance (2-sided) | Exact Sig. (2-sided) | Exact Sig. (1-sided) |
|---|---|---|---|---|---|
| Pearson Chi-Square | 56,190[a] | 1 | < .001 | | |
| Continuity Correction[b] | 55,878 | 1 | < .001 | | |
| Likelihood Ratio | 55,516 | 1 | < .001 | | |
| Fisher's Exact Test | | | | < .001 | < .001 |
| Phi | -.01 | | < .001 | | |
| Cramer's V | .01 | | < .001 | | |
| N of Valid Cases | 726795 | | | | |

a. 0 cells (0.0%) have expected count less than 5. The minimum expected count is 998.25.

b. Computed only for a 2x2 table

**Table 5. Temporal present and atemporal present crosstabulation.**

| | | | Atemporal Present | Temporal Present | Total |
|---|---|---|---|---|---|
| Platform | Twitter | Count | 642406 | 182300 | 824706 |
| | | Expected Count | 584205.0 | 240501.0 | 824706.0 |
| | | % within Platform | 77.9% | 22.1% | 100.0% |
| | | Adjusted Residual | 443.3 | 443.3 | |
| | Welbo | Count | 7548 | 85268 | 92816 |
| | | Expected Count | 65749.0 | 27067.0 | 92816.0 |
| | | % within Platform | 8.1% | 91.9% | 100.0% |
| | | Adjusted Residual | 443.3 | 443.3 | |
| Total | | Count | 649954 | 267568 | 917522 |
| | | Expected Count | 649954.0 | 267568.0 | 917522.0 |
| | | % within Platform | 70.8% | 292% | 100.0% |

Similarly, recognizing the prevalence of atemporal present on English Twitter suggests that focusing on factual statements and logical reasoning may resonate better with Western users. Tailoring communication approaches based on temporal orientations can improve intercultural understanding and engagement.

## Conclusion

Psychological literature about cross-cultural differences in temporal orientation has presented contradicting findings on how North American and East Asian cultures differ in their focus on past, present, and future. The big data approach we took addresses the discrepancies and methodological challenges from previous research. Our results highlight the atemporal present category of temporal orientation, suggesting an additional dimension to consider for time-related studies. Despite some common limitations of using social media data (i.e., sampling bias) and natural language processing (i.e., coding psychological constructs through language), analyzing large sets of social media posts directly from local users provides a viable and economical way to conduct cross-cultural research while ensuring measurement equivalence and ecological validity. Furthermore, using behavior measures–language use–complements previous self-report measures of temporal orientation, ensuring construct validity and clarity.

More broadly, our study shows the value of merging multiple disciplines–psychology, computer science, and linguistics–in one approach to answering long-debated questions that were previously studied within one domain of discipline. The advent of machine learning and the

**Table 6. Chi-square test results for atemporal and temporal present orientations.**

| | Value | df | Asymptotic Significance (2sided) | Exact Sig. (2 sided) | Exact Sig. (1 sided) |
|---|---|---|---|---|---|
| Pearson Chi-Square | 196549.15[a] | 1 | .000 | | |
| Continuity Correction[b] | 196545.78 | 1 | .000 | | |
| Likelihood Ratio | 184007.74 | 1 | .000 | | |
| Fisher's Exact Test | | | | .000 | .000 |
| Phi | .46 | | .000 | | |
| Cramer's V | .46 | | .000 | | |
| N ofvalid Cases | 917522 | | | | |

a. 0 cells (0.0%) have expected count less than 5. The minimum expected count is 27067.03.

b. Computed only for a 2x2 table

exponential growth of social media data present unprecedented opportunities to delve deeper into the intricacies of cross-cultural differences. By harnessing these technological advancements, researchers, social scientists, and organizations can gain invaluable insights that were previously unattainable through traditional methods. These insights can facilitate a more comprehensive understanding of cross-cultural differences, paving the way for enhanced intercultural communication, global collaboration, and the development of targeted strategies in areas such as marketing, education, and policy-making.

## Supporting information

**S1 Table. Previous sampling examples.**
(DOCX)

## Acknowledgments

The authors would like to acknowledge Jialin Wang and Hanyu Sun for their contributions on the data collection process and Fanyi Zhang for her contributions to the data annotation process.

## Author Contributions

**Conceptualization:** Xin Daphne Hou, Lyle Ungar, Louis Tay.

**Data curation:** Young-Min Cho, Tingdan Zhang, Mingyang Li.

**Formal analysis:** Xin Daphne Hou, Young-Min Cho, Garrick Sherman, Mingyang Li.

**Investigation:** Sharath Chandra Guntuku, Garrick Sherman, Tingdan Zhang, Mingyang Li, Lyle Ungar, Louis Tay.

**Methodology:** Sharath Chandra Guntuku, Lyle Ungar, Louis Tay.

**Project administration:** Xin Daphne Hou.

**Resources:** Sharath Chandra Guntuku, Lyle Ungar, Louis Tay.

**Software:** Sharath Chandra Guntuku, Garrick Sherman.

**Supervision:** Sharath Chandra Guntuku, Lyle Ungar, Louis Tay.

**Visualization:** Xin Daphne Hou, Sharath Chandra Guntuku, Young-Min Cho, Garrick Sherman.

**Writing – original draft:** Xin Daphne Hou, Young-Min Cho, Louis Tay.

**Writing – review & editing:** Xin Daphne Hou, Sharath Chandra Guntuku, Young-Min Cho, Garrick Sherman, Tingdan Zhang, Lyle Ungar, Louis Tay.

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
