## [Decision Letter · Decision Letter 0]

10 May 2023

PONE-D-23-03984A Cross-cultural Examination of Temporal Orientation Through Everyday Language on Social MediaPLOS ONE

Dear Dr. Hou,

Thank you for submitting your manuscript to PLOS ONE. After careful consideration, we feel that it has merit but does not fully meet PLOS ONE’s publication criteria as it currently stands. Therefore, we invite you to submit a revised version of the manuscript that addresses the points raised during the review process.

We look forward to receiving your revised manuscript.

Kind regards,

Tinggui Chen

Academic Editor

PLOS ONE

Journal Requirements:

2. a. For studies reporting research involving human participants, PLOS ONE requires authors to confirm that this specific study was reviewed and approved by an institutional review board (ethics committee) before the study began. Please provide the specific name of the ethics committee/IRB that approved your study, or explain why you did not seek approval in this case.

b. Please provide additional details regarding participant consent. In the ethics statement in the Methods and online submission information, please ensure that you have specified what type you obtained (for instance, written or verbal, and if verbal, how it was documented and witnessed). If your study included minors, state whether you obtained consent from parents or guardians. If the need for consent was waived by the ethics committee, please include this information.

Additional Editor Comments:

I have completed my evaluation of your manuscript. The reviewers recommend reconsideration of your manuscript following major revision. I invite you to resubmit your manuscript after addressing the comments below.

Reviewers' comments:

Reviewer's Responses to Questions

**Comments to the Author**

1. Is the manuscript technically sound, and do the data support the conclusions?

Reviewer #1: Yes

Reviewer #2: Yes

2. Has the statistical analysis been performed appropriately and rigorously? 

Reviewer #1: Yes

Reviewer #2: Yes

3. Have the authors made all data underlying the findings in their manuscript fully available?

Reviewer #1: No

Reviewer #2: Yes

4. Is the manuscript presented in an intelligible fashion and written in standard English?

Reviewer #1: Yes

Reviewer #2: Yes

5. Review Comments to the Author

Reviewer #1: The article presented is important and useful in the contemporary dispensation. There are however a few consideration that the authors need to pay attention to inorder to improve the quality of the manuscript. Please find attached a report of the reviewer comments attached. This should be made available to the authors for them to consider improving the quality of the manuscript.

Reviewer #2: The manuscript discusses an interesting issue of global importance; A Cross-cultural Examination of Temporal Orientation Through Everyday Language on Social Media, while the manuscript is well delivered. I provide some comments for improvement.

The sentence ‘Implications of the findings are discussed’ should remove from the abstract.

Statement of the research problems need a more clear articulation.

Can you please provide the equation of the logistic regression classification model and Randomized Principle Component Aanalysis with empirical model?

Please add a bit more the main findings.

6. PLOS authors have the option to publish the peer review history of their article (what does this mean?). If published, this will include your full peer review and any attached files.

Reviewer #1: **Yes: **Daniel Adu Ankrah

Reviewer #2: **Yes: **Mst. Esmat Ara Begum

---

## [Author Response · Author response to Decision Letter 0]

27 Jun 2023

Dear Dr. Chen, 

We appreciate the opportunity to revise the paper and are very grateful for the helpful feedback you and the reviewers provided. We have made every effort to address the concerns raised and incorporate helpful suggestions in our paper. As a result, we believe that the paper is considerably better due to the constructive feedback. In the followings, we provide detailed responses to all the reviewer concerns and believe we have addressed them.

Abstract

Reviewer 1 has suggested to remove the sentence “Implications of the findings are discussed’ should be removed from the abstract”. 

• We have eliminated this sentence from the abstract. 

Reviewer 2 has suggested spelling out “United States of America” in full for the first time in the abstract. 

• We have spelled it out in full. 

Reviewer 2 has suggested that it is essential to indicate the implications of the findings in the abstract and offer a policy recommendation.

• Please see our revision at the bottom of the Abstract section on page 2. 

Introduction

Review 1 has suggested that statement of the research problems needs a clearer articulation.

• Please see the last paragraph of Introduction/bottom paragraph on page 5. We added a sentence to clearly define the research problem and objective. 

Method

Both reviewers 1 and 2 have asked to “provide the equation and output of the logistic regression classification model and Randomized Principal Component Analysis with empirical model”

• We added the equation in the last paragraph of the Method section on page 9 and additional examples of other studies that have followed similar methodology approaches in the past. 

Review 2 has suggested that under the method section, it is important for the authors to state the research design, whether it is a mixed method or a qualitative study.

• We have clarified the language in the beginning of Method section to be “mixed method”. 

Discussion

Review 1 has suggested to add a bit more the main findings.

• We have re-iterated the main findings at the start of the Discussion section. 

Reviewer 2 has suggested that previous studies have generally posited temporal differences in past and future between Eastern and Western cultures. The authors need to reference these previous studies.

• We have referenced these previous studies in the manuscript. 

Conclusion

Reviewer 2 has suggested a need to highlight the implications of the findings for culture and social media influence.

• Please see the revision in the last paragraph of the Discussion section/on the bottom paragraph of page 15. 

Reviewer 2 has suggested a need to present a conceptual or theoretical framework to guide the study.

• We appreciate this feedback. As this paper is an empirical test of the temporal orientation using a new methodology (i.e., social media language), there is no strong conceptual or theoretical framework. Rather, our paper is driven by the phenomenology of temporal orientation. As mentioned in the abstract, this is “an inductive approach” rather than a top-down theoretical deductive approach. We have also emphasized this in our introduction and in the discussion too. Finally, in our discussion, we have pointed out how this study contributes to the current trichotomous framework of temporal orientation and call for an expanded view.

We again thank you and the reviewers for the helpful feedback. In the above, we provide detailed responses to all the reviewer concerns and believe we have addressed them. However, if you should feel that additional issues are needed, we would be happy to accommodate or provide more information.

Sincerely,

Daphne Hou

---

## [Decision Letter · Decision Letter 1]

3 Oct 2023

A cross-cultural examination of temporal orientation through everyday language on social media

PONE-D-23-03984R1

Dear Dr. Hou,

We’re pleased to inform you that your manuscript has been judged scientifically suitable for publication and will be formally accepted for publication once it meets all outstanding technical requirements.

Kind regards,

Tinggui Chen

Academic Editor

PLOS ONE

Additional Editor Comments (optional):

Reviewers' comments:

Reviewer's Responses to Questions

**Comments to the Author**

1. If the authors have adequately addressed your comments raised in a previous round of review and you feel that this manuscript is now acceptable for publication, you may indicate that here to bypass the “Comments to the Author” section, enter your conflict of interest statement in the “Confidential to Editor” section, and submit your "Accept" recommendation.

Reviewer #1: All comments have been addressed

Reviewer #2: All comments have been addressed

2. Is the manuscript technically sound, and do the data support the conclusions?

Reviewer #1: Yes

Reviewer #2: (No Response)

3. Has the statistical analysis been performed appropriately and rigorously? 

Reviewer #1: Yes

Reviewer #2: (No Response)

4. Have the authors made all data underlying the findings in their manuscript fully available?

Reviewer #1: Yes

Reviewer #2: Yes

5. Is the manuscript presented in an intelligible fashion and written in standard English?

Reviewer #1: Yes

Reviewer #2: Yes

6. Review Comments to the Author

Reviewer #1: All comments raised have been adequately addressed. The authors have painstakingly addressed the comments and even gone beyond to improve the manuscript. I recommend that the manuscript can now be accepted for publication. Thank You.

Reviewer #2: Authors have made great effort to the betterment of the manuscript. It has desire to the public and the literature in terms of new communication technology and methodological approach. I recommend for getting it published if it fit the journal scope.

7. PLOS authors have the option to publish the peer review history of their article (what does this mean?). If published, this will include your full peer review and any attached files.

Reviewer #1: **Yes: **Daniel Adu Ankrah

Reviewer #2: **Yes: **Mst. Esmat Ara Begum, PhD

---

## [Editor Report · Acceptance letter]

30 Nov 2023

PONE-D-23-03984R1 

A cross-cultural examination of temporal orientation through everyday language on social media 

Dear Dr. Hou:

I'm pleased to inform you that your manuscript has been deemed suitable for publication in PLOS ONE. Congratulations! Your manuscript is now with our production department. 

Kind regards, 

on behalf of

Dr. Tinggui Chen 

Academic Editor

PLOS ONE